# Deep Reinforcement Learning based Insight Selection Policy

## Abstract

We live in the era of ubiquitous sensing and computing. More and more data is being collected and processed from devices, sensors, and systems. This opens up opportunities to discover patterns from these data that could help in gaining better understanding into the source that produces them. This is useful in a wide range of domains, especially in the area of personal health, in which such knowledge could help in allowing users to comprehend their behavior and indirectly improve their lifestyle. Insight generators are systems that identify such patterns and verbalize them in a readable text format, referred to as insights. The selection of insights is done using a scoring algorithm which aims at optimizing this process based on multiple objectives, e.g., factual correctness, usefulness, and interestingness of insights. In this paper, we propose a novel Reinforcement Learning (RL) framework that for the first time recommends health insights in a dynamic environment based on user feedback and their lifestyle quality estimates. With the use of highly reusable and simple principles of automatic user simulation based on real data, we demonstrate in this preliminary study that the RL solution may improve the selection of insights towards multiple pre-defined objectives.

## 1 Introduction

The latest developments in big data, internet of things and personal health monitoring have led to the massive increase in the ease and scale at which data has been collected and processed. Learning from the information present in the data has shown to help to gain wisdom to better run businesses, manage health care services and even maintain a healthier lifestyle. Such understanding are mostly in the form of identifying significant rise or fall of a certain measurement given a context of interest. Let's say that the sleep data logs of a user of a health monitoring service shows that the time at which they went to sleep was later during the weekends in comparison to weekdays. This can be informed to the user as a statement such as, "You went to sleep later during the weekends than the weekdays". Here, the time at which they went to sleep is the measurement and the fact of the day being a weekday or a weekend is the context of interest. We call such statements as 'insights'. Providing such insights that accurately describe the scenarios during which certain health parameter improved or deteriorated could enable the user to make better lifestyle choices. Moreover, it has been accepted Abraham & Michie (2008) that providing relevant information to the user could improve their behavior.

The insight generation task can be seen as a natural language generation task where a generator model creates appropriate insight statements. A generalized framework for such an insight generator (Genf) was proposed, in which components to analyze the data and generate the statements played an important role (Susaiyah et al., 2020). More importantly, the framework has a provision to capture user feedback mechanism that understands what type of insights they are interested in. Implementations of this framework have shown to incorporate the "overgenerate and rank" approach, in which all possible candidates as per definition are generated and later filtered using a calculated rank or a score (Gatt & Krahmer, 2018; Varges & Mellish, 2010).

The selection of the most relevant insight via ranking or scoring from a list of multiple insights is an ongoing research topic. Earlier works have utilized purely statistical insight selection mechanisms where the top ranking insights based on a statistical algorithm are selected (Härmä & Helaoui, 2016), often combined with machine-readable knowledge (Musto et al., 2017). Other approaches

combined neural networks with the knowledge of statistical algorithms with simulated user feedback (Susaiyah et al., 2021). All the above techniques have limitations with respect to over-simplification of user-preference or need for a huge amount of data.

On the other hand, as noted in Afsar et al. (2021), the very nature of a recommendation is a sequential decision problem thus that could be modelled as a Markov Decision Process (MDP). Reinforcement Learning (RL) can therefore be used to solve this problem, taking into account the dynamics of the user's interactions with the system, its long-term engagement to specific topics and more complex feedbacks than binary ratings.

In this paper, we introduce a novel Deep Reinforcement Learning (DRL) based framework to recommend health insights to users based on their health status and feedback. While it incorporates previously developed insights generation techniques (Susaiyah et al., 2020), the presented framework is based on a completely new training pipeline that uses real-time simulated data instead of retrospective data and an objective of choosing the best insight instead of assigning scores to all insights. By the use of DRL, the presented system is able to deliver both useful and preferable insights. To the best of our knowledge, there is no other existing system capable of reaching both of those objectives. We evaluate it in this preliminary study in terms of significance of life quality improvements, speed and accuracy of adaptation to the dynamics of user preferences, and deploy-ability in a real life scenario.

## 2 RELATED WORK

Traditionally, insights were generated using associate rule mining techniques (Agrawal & Shafer, 1996), where associations between different contexts in a dataset are discovered. However, they do not work for continuous variables. This led to the work of Härmä & Helaoui (2016) where both continuous and categorical variables were considered. However, it lacked the ability to adapt to specific users, which is very important as what we consider as an insight is highly subjective. Later, a Genf was introduced in Susaiyah et al. (2020) to incorporate the users as part of the insight generation system. This framework requires highly dynamic mechanisms to rank and recommend the insights based on the dynamics of user interests.

To have a clear understanding of the main goals of this task, the survey Pramod & Bafna (2022) summarizes and presents ten challenges to overcome for conversational recommender systems. As one of them, our approach was designed to respond to nine out of the ten, the latest being only related to dialogue management, which is not part of the scope of the present study. The main challenges that we focus on are to: 1) keep the reliability in the ratings given by the user, 2) minimize the time spent for rating, 3) allow cold start recommendations, 4) balance cognitive load against user control, 5) remove user's need to tell the technical requirement, 6) allow a fallback recommendation when no items found, 7) limit item presentation complexity, 8) allow domain-dependent recommendations using a Genf, and 9) convince users about the recommendation by presenting facts.

The neural insight selection model presented in Susaiyah et al. (2021) was agnostic to the overall objective of the user to use the insights: to improve a behavior/performance. The authors modeled the problem as a scoring objective that assigns a score between 0 and 1 on how relevant it is to the user. Tops insights were selected on need basis in order to improve the systems understanding of users preference. The main drawback of this approach is that it only focuses on insight selection for user preferences and used supervised learning from binary feedback. Therefore, it could neither consider the long term nor short term impact a given insight will produce for a given user. Nowadays, the problem of long time interaction, understand daily recommendation over multiple months, can be solved using DRL. However, DRL is known to be a very consuming approach in terms of sample efficiency whether being model-free as policy-gradient, value-based, actor critic or model-based. All DRL algorithms as SAC (Haarnoja et al., 2018), A3C (Mnih et al., 2016), DDPG (Lillicrap et al., 2015), DQN (Mnih et al., 2013) or PPO (Schulman et al., 2017) suffer from this problem and require, on average, several millions of interactions with their environment to solve complex problems as demonstrated in their paper. This is even more problematic as the continuous supervised learning in Susaiyah et al. (2021) already required on average 15.6 labelled insights with the user feedback every day.

To implement such a DRL approach in the healthcare domain, several challenges need to be solved as described in Riachi et al. (2021). Usually data needs to be obtained through time-consuming, and not necessarily realistic, studies requiring the patient to wear electronic devices. Therefore, this leads to the usage of limited observational data, which denies the freedom to explore different interaction strategies during the training process. As a solution to solve this issue in DRL, dialogue (Zhao et al., 2021) or video game (Young et al., 2018; Yang et al., 2020) approaches, rely on episodic memory to exploit the observations in order to optimize both the training and decision processes. Researchers also worked on other strategies such as designing simulations from the data itself and plenty of examples can be found in the literature as HeartPole and GraphSim (Liventsev et al., 2021), or the dozens of RL applications presented in Yu et al. (2019). Another major challenge, as detailed in Riachi et al. (2021), is the design of the most appropriate reward function. To this end, RL solutions often exploit counterfactuals available in their own collected data, or datasets such as MIMIC-III (Johnson et al., 2016), to design their environment. Indeed, knowing what would have happened taking another decision allows for a more efficient reward function and training, but requires a careful and manual design. This conception, then, makes it difficult to reuse the ideas from one specific problem to another.

In an objective of simplifying the reusability of a such DRL approach, the authors focused on developing an environment in which the selection process is given an unlimited amount of data to learn. To leverage the limitations about the data, this work presents a generic replication procedure for individuals' behaviors from existing data, requiring only start and stop times of activities allowing to automatically create a simulated user. Even though the selection problem is represented as an MDP by the DRL policy, this user simulator was designed as a Gaussian-based probabilistic state machine. From the use of this state machine, the policy network can learn through trial and error on how its actions impact the simulation, based on very simple assumptions, without the need of counterfactuals in the original dataset. Moreover, the probabilistic nature of the simulation generates an infinite, yet realistic amount of data, therefore not requiring manually designed reward functions to guide the policy during its training.

The approach detailed in Susaiyah et al. (2021) proposed to use an insight generator with a neural network to select insights to be given to the user. Depending on the simulated feedback, the system was able to change the selected insights and was therefore demonstrated to be robust to the user interests when tested with preferences variations. The role of this preliminary study is to show the interest of using DRL instead of supervised learning, requiring a continuous training on the current preferences of the user. Moreover, the reward system of RL allows taking into account more complex metrics of performance than just binary preferences (e.g., life quality improvements such as sleep quality). The experiments presented here show the robustness of this approach, considering the variability and complexity of the simulated user's behavior and associated preferences. In order to give to the reader the easiest explanation of our approach, we use the same sleeping simulation example throughout this paper.

## 3 METHODOLOGY

We designed a complete training pipeline, presented in Figure 1, composed of 3 stages that are repeated every day: (1) insight generation, (2) insight selection, (3) user lifestyle simulation. The pipeline was designed to simulate the interaction between the policy network and a user, aiming at two distinct objectives being to select insights that are appreciated by the user and beneficial to their life quality.

### 3.1 INSIGHT GENERATION

Insight generation is performed using the insight generator described in Susaiyah et al. (2021). The proper generation of insights is based on predefined schemas of comparison such as

$$sleep\_period : 1 \quad measurement \quad sleep\_period : 2 \quad mean : 2$$
$$short\_period : 1 \quad measurement \quad measurement\_benchmark : 2$$

with :1 and :2 respectively referring to subset 1 and 2 that are compared. In the above example, the first schema states that all insights that compare a given measurement across two subsets of time-periods like months, weeks, days of week, etc. A example from this schema would

be, "You sleep less on Mondays than the other days". The second schema compares a measurement during a period with a predefined benchmark value such as 8 hours of sleep per night or time of going to bed at around 11PM. For example, such an insights would be: "You sleep less than 8 hours on Mondays". The queries to extract the relevant data subsets are routinely called by the system to validate these insights. This approach allows for a very precise control over the generation process, encoding all the possible comparisons between two subsets of the data.

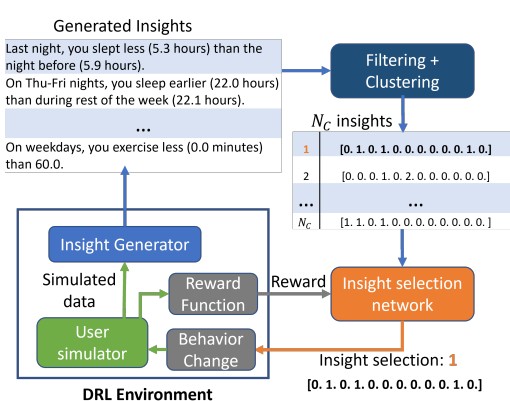

Figure 1: Representation of the daily insight selection pipeline.

From those comparisons, a Kolmogorov-Smirnov test is computed on the two subsets of all insights to score their individual statistical significance. The non-significant insights, with a $p_{value} < 0.05$, are filtered and removed. Subsequently, a score for relevance ($Score_F$) is calculated, determined by both the completeness of the data and the tolerance level as shown in Equation 1. The first factor is calculated using the expected rate of sampling of the data $F_{exp}$ that is preset manually depending upon the data source, the time period of the queried data $T$, and the number of data samples available $N_{rec}$. The second factor, is calculated using $\gamma$, a weighting factor that determines the slope of the sigmoid function and $\delta$, the difference in the means of the measurements across the two contexts.

$$Score_F = \frac{N_{rec}}{F_{exp} * T} * \frac{1}{1 + \exp(-\frac{\gamma\delta}{\tau})} \quad (1)$$

Each insight is assigned a feature vector of size $N_T$ based on the bag of words (BoW) embedding. This encodes, into a vector format, the number of occurrences of each word of a predefined dictionary of words present in the insight. Finally, K-means clustering is performed on the feature vectors using Euclidean distances. Only the insights having the highest relevance scores are chosen forward from each of the K clusters.

### 3.2 INSIGHT SELECTION

From the feature vectors of the available insights after generation, the policy network was trained to directly select one of the $N_C$ insights that are composed of the K-clustered insights and $N_C - K$ benchmark insights, ensuring the policy network to always have comparison with benchmark values for each measurement.

Finally, to simulate the everyday lifestyle of a user, we opted for a state machine with probabilistic and time dependent transitions. Depending on the insight that is selected by the policy network, the behavior of the user is assumed to tend towards the optimal behavior implied by the insight. For example, an insight as 'You sleep around 3 hours later than 21:00 on weekdays' implies to the user to sleep earlier. A new day of simulation is then computed and new data about the user (e.g., sleep duration, exercise duration, etc.) is generated. If the policy is trained on user satisfaction, it will be rewarded by a discrete feedback whether the topics from the selected insight meets the interests of the user or not. Otherwise, the policy network is rewarded using metrics evaluating the life quality of the user.

### 3.3 USER SIMULATION

In an effort to be able to model any kind of time dependent behavior, our system relies only on three types of input data for each activity the user might be doing, i.e., the day, the start time and the stop time on which it happened. From this information, a state machine is automatically constructed with one state representing each activity. Additionally, one other state, named Idle, is created to play the role of link between all the others. In this state machine, each transition between two states is modelled as a Bayesian Gaussian Mixture Model (GMM) (Roberts et al., 1998; Blei & Jordan, 2006; Attias, 2000) in order to preserve the random nature of human behavior depending on the current time and day in the simulation. An example of a state machine with seven activities is presented

in Figure 2. As the reader can note, all the possible activities are linked to the state Idle by one transition in each direction.

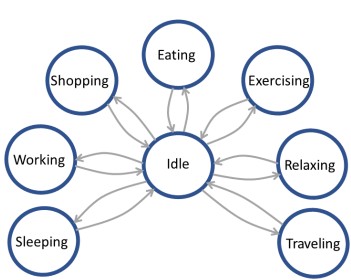
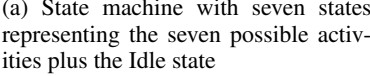

(a) State machine with seven states representing the seven possible activities plus the Idle state

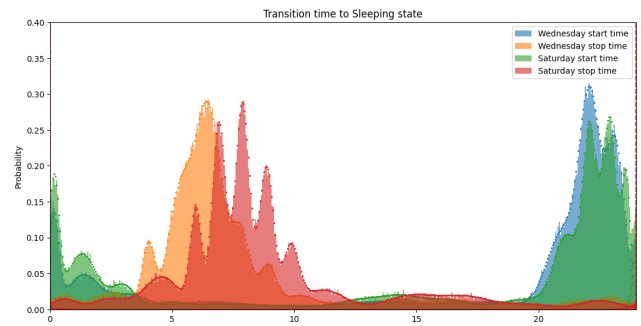

(b) Associated probabilities to Sleeping state transitions for Wednesday and Saturday in function of the time of the day

Figure 2: Example of state machine and associated probabilities

### 3.3.1 DATA EXTRACTION

In order to construct this state machine, the process is composed of three steps: (1) data filtering that depends on the activities considered and extraction of features such as start and stop time or duration, (2) Gaussian estimation with optional iterative component reduction and (3) creation of a new state in the state machine. As presented in the algorithm 1, two Bayesian GMMs are computed for every day of the week and every activity to simulate. First, the start times, stop times and durations related to a specific day and activity is filtered. Then, a Bayesian GMM is computed, using the library Scikit-Learn (Pedregosa et al., 2011), from the filtered start times, and another one from the filtered stop times of the activity. An example of probabilities associated with the activation of the state Sleeping is given in Figure 2 for two different days of the week after the Bayesian GMM estimations. In this example, we can observe that the wake time is on average later and more scattered for Saturday than for Wednesday. The initial number of component for the GMM was set to 24 and is decreased until a mixture model can be computed from the data. This value has been empirically selected and is suited for the later experiments presented in this paper, but may need to be modified. In the case where one considered activity would be happening consistently more than 24 times a day, a higher number of Gaussians would be needed to simulate it accurately. Finally, the Bayesian GMM corresponding to the start times of the activity is added to the transition from the state Idle of the state machine and the other GMM added to the transition to the state Idle.

Additionally, two other constraints are added to the transitions: the horizon and the proportion of the activity during the day. The horizon constraint is a mechanism of attention that is used here to filter the sampled points from the GMM and ensures that (1) transition from Idle to any other state is selected within a range of 3.5 hours and (2) transition from a given state back to Idle happens within the maximum duration of the activity observed in the data. The proportion constraint is used to balance the activation likelihood of all states by normalizing them by their number of occurrences in a given day. Indeed, the GMM only provides a time-dependent activation probability, independent of other states. For example, incorporating the proportion of Working as a constraint, the simulator knows that the likelihood of the occurrence of Working will be much greater than Shopping on the weekdays contrary to the weekend.

### 3.3.2 RUN TIME EXECUTION

On run time, the simulation starts on Monday at 00:00 and is incremented by one minute at each new step. The first state to be activated is Idle and the following process, represented in Figure 3, is repeated for each new active state. To illustrate how a transition is selected, we will consider the state machine in Figure 2, the current active state Idle and the current time 12:00. For all the possible transition from Idle (i.e., to Sleeping, Working, Shopping, Eating, Exercising, Relaxing or Travelling), the transition processing in Figure 3 is repeated.

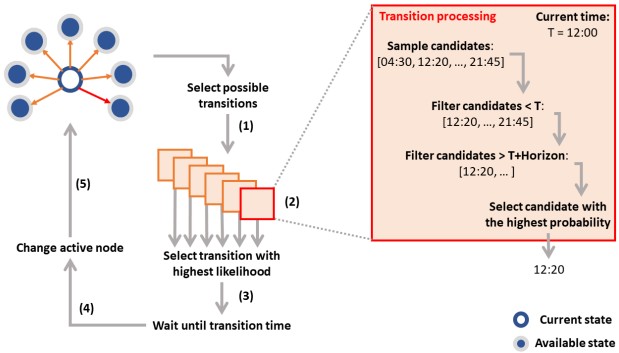

Figure 3: Representation of the state machine run time process.

The first step is to sample 100 transition candidates, randomly generated from the GMM associated to the transition, as 12:20, 4:30 or 21:45 in this example. Then, the candidates referring to transitions in the past are removed, such as 4:30. The third step is to remove candidate times further than the horizon constraint associated to the transition, here it corresponds to 3.5 hours, filtering 21:45. Finally, the transition with the highest probability is selected from the remaining candidates. Once this process has been repeated for every possible transition, the one wielding the highest $likelihood = probability * proportion$ is selected. The simulation then advances until the transition time is reached. The new state is activated, and the loop starts again. In case no activation is found, which happens by the end of the day as there are fewer candidates, the simulation goes to the next time step.

### 3.4 REINFORCEMENT LEARNING

#### 3.4.1 OBSERVATIONS

However, to take its decisions, the policy network is not able to see the underlying state machine representing the user nor the history of activated states. Instead, it is able to see both the history of selected insights for the last $T$ days, with their impact on the user (represented by life quality metrics) and the currently available insights as presented is figure 4. As each day, the policy network selects one new insight, one day is represented by a matrix with as many rows as the features dimension and 2 columns. The feature vector of the selected insight is represented ver-

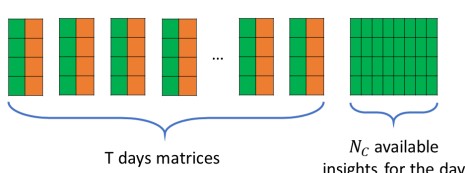

Figure 4: Observation matrix

tically while the second column contains information about the metrics to be optimized, for a given day. In our example, these values are related to exercise duration, feedback about the selected insight or sleep quality measures. When a new day of data is available, the oldest day matrix is removed, the T days matrices are rolled to the left, the new day matrix is added to the right, and the $N_C$ available insights are updated.

#### 3.4.2 ACTIONS

We assume that the users improve their behavior when presented with relevant insights as mentioned in Abraham & Michie (2008). For this, we define a $r_{value}$ to be the difference between the predefined recommended value and the current value of a measurement. This factor helps to simulate behavior change based on the insight presented to the user using Equation 2.

$$new\_value = old\_value + \beta * r_{value} \tag{2}$$

. Where $\beta$ is a lifestyle improvement factor. For example, if an insights says that the user generally sleeps 5 hours instead of 8 hours, the sleep will be modified to 5.3 hours with $\beta = 0.1$. Further, to facilitate this, the Gaussian means $\mu$ are modified in the state machine by equation 3.

$$\mu = \beta * r_{value} * \mu \tag{3}$$

#### 3.4.3 REWARDS

For this study, we decided that the reward should translate (1) the improvements of the sleep quality, (2) the exercise duration and (3) satisfaction of the user's interests by giving them the insights they

want to see. In order to give a realistic sense of the user's sleep quality, we decided to use the Pittsburgh Sleep Quality Index or PSQI (Buysse et al., 1989). The PSQI is composed of 19 questions grouped into seven components respectively related to subjective sleep quality, sleep latency, sleep duration, habitual sleep efficiency, sleep disturbances, use of sleeping medications and daytime dysfunction. Each of these components are equally rated from 0 to 3 and summed to create a global PSQI score, therefore ranging from 0 to 21. In this study, the simulation being rather simple on purpose, only sleep duration and habitual sleep efficiency have been computed from the simulation and the other components were assigned a constant value. This results in a global PSQI score ranging from 4 to 10, the lower being the better. The exercise duration of the user is computed as the exercise done in one week. Given our study example presented in figures 2, exercise can only happen on the Tuesday, Saturday and Sunday, raising the optimal value to 180 minutes of exercise per week.

Finally, to incite the policy network to give insights that the user appreciates, we decided of four different topics that could be selected as of interest for the user: sleep time, wake time, sleep duration or exercise. At each epoch of training, a new order of those four topics is randomly generated. Every three weeks, the topic of interest is modified following the previously generated order. If the selected insight contains the current topic of interest, the reward can very simply be 1 and otherwise 0.

## 4 EXPERIMENTS

In order to create a digital twin of a real user, we used the American Time Use Survey 2003-2020 dataset. This dataset contains information about how people spend their time in a day of 24 hours, each household being interviewed only once. From the 219,368 individuals interviewed, we focused on the working population (21 to 50 years old) which represents 68 percent of the respondents. Among the hundred of available activities, we only considered seven of them: sleeping, working, eating, exercising, relaxing, shopping and travelling. We intentionally created a user with a difficult sleep, as he/she is waking up in the middle of the night and also very early in the morning by taking into account thousands of persons with very different lifestyles as it is the perfect example for the policy network to work on. Indeed, the resulting simulated user has a great potential for improvements, generating a lot of very relevant recommendation insights.

For the following experiments, the features encode 15 possible topics present in an insight: measurement benchmark, exercise duration, sleep duration, weekday, wake time, sleep time, weekend, Monday, Tuesday, Wednesday, Thursday, Friday, Saturday, or Sunday. These topics have been selected in order to inform the policy network about: the measurement conveyed in the insight, if it compares it to a benchmark value or to which day of the week it refers to.

For both topic selection and insight selection objectives, we decided to go for the same number of choices. That is, the number of topics ($N_T$) and the number of insights to select from ($N_C$) are both equal to 15. In the case of insight selection, 4 out of the 15 insights are reserved for benchmark insights, which ensure that the policy network has enough choices at all times. For the training of the policy network, the impact an insight can have on the simulation was entirely described in the previous sections. For the test part presented in the next sections, we set an even more difficult problem by dividing by two the lifestyle improvement factors and by ten the lifestyle fallback factors. The values of interest available to the policy network every day were the PSQI with its seven components, the exercise duration and the feedback from the user. The policy network had access to the history of the past 7 days, which means $T = 7$.

### 4.1 INSIGHT SELECTION FOR HEALTHCARE

For the first experiment, the reward was based on the PSQI and the exercise duration per week. In order to evaluate the capacity of the policy to learn relevant selection behaviors, the training was repeated with each of the following reward functions:

$$r_{PSQI} = \frac{-PSQI + 10}{6} \tag{4}$$

$$r_{EXE} = sigmoid(\frac{T_{EXE}}{180}) \tag{5}$$

$$r_{FULL} = \frac{-PSQI + 10}{6} + sigmoid(\frac{T_{EXE}}{180}) \tag{6}$$

$$r_{FULL\_LIN} = \frac{-PSQI + 10}{6} + \frac{T_{EXE}}{180} \qquad (7)$$

Moreover, to have a point of comparison, we performed two other tests by randomly selecting an insight every day (i.e., RANDOM) and not selecting any insights at all (i.e., BASELINE). Each epoch of RL was composed of 3 weeks of only simulation plus 21 weeks of interaction, which means 147 actions required. As the process is time-consuming and this study is a preliminary study, the policy network was trained only for 500,000 steps and required an accelerated training process to be executed in a reasonable amount of time. For this reason, during the training, insights were only computed once every week instead of every day.

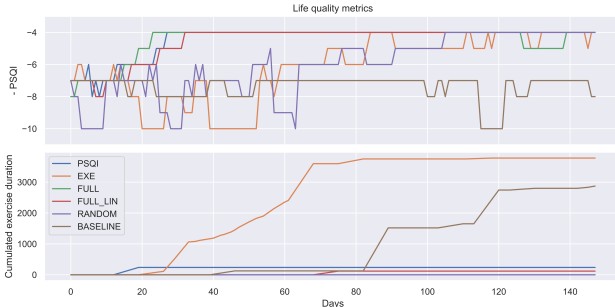

Figure 5: User's life quality over one test simulation with different insights selection policies.

In the figure 5 is presented how the life quality of the simulated user evolve over time with the different selection policies. On the upper plot, three policies are able to increase the PSQI value within 35 days up to 4: the PSQI, FULL and FULL_LIN policies. The other policies, EXE and RANDOM, are then able to reach the same value within 120 days, while the BASELINE stays around 7. On the lower plot, the EXE policy is able to make the user perform almost 4000 hours of exercise over the simulation, BASELINE is almost reaching 3000 hours as exercise can happen without any intervention from the policy while FULL, RANDOM and PSQI are not getting more than 500 hours of exercise. Additional results can be found in the Appendix.

All policies using the PSQI in their reward: the PSQI, FULL and FULL_LIN policies, were the fastest to improve the sleep quality of the user compared to EXE, RANDOM or BASELINE. This highlights the capacity of the policy network to learn the optimal behavior in order to improve sleep quality by different strategies. The EXE policy was the one pushing the user to do the most exercise, thus that learned how to maximize this behavior, while also selecting other types of insights as the PSQI can be seen improving. However, the policies that were trained to optimize both PSQI and exercise duration failed on the later objective. This could be due to an improper weightage of the two components of the reward, or a lack of training time.

### 4.2 INSIGHT SELECTION WITH FEEDBACK

The problem explained in 4.1 only partially addresses the preference of the user. For example, if the user doesn't improve a particular aspect of their lifestyle in spite of repeatedly informing about it, the model automatically realized its uselessness and discards such insights. However, some insights can get repetitive over time and the user might want to start to see other types of insights. For this, we carry off the second experiment, where we focused on selecting insights relevant to the preference of the user rather than lifestyle scores. As the reward is solely composed of the feedback of the user, that is generated at the start of the epoch, it does not require the state machine to run at every step from then. Therefore, each epoch of RL was composed of 6 weeks of simulation only, in order to generate enough insight candidates for

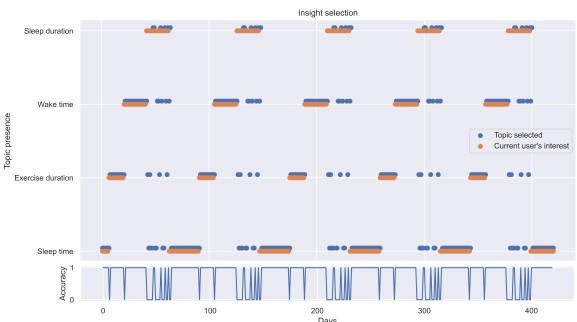

Figure 6: Comparison of user's topic of interest and the presence of it in the selected insight by the policy over time.

every topic, and 60 weeks of interaction. This process allowed to greatly decrease the computing time, dividing it by a tenth.

In Figure 6 we present the behavior of the policy network after 12 million steps of training. The policy network learned (1) that only 4 topics were of interest for this user and (2) to keep proposing the same type of insights while the feedback of the user is positive. It is perfectly able to select the right insight when the topic of interest is either sleep time, exercise duration or wake time. When the topic of interest is sleep duration, the policy does not know which insight to select, and it results in choosing insights on other topics. This probably results from a lack of training time.

## 5 DISCUSSION

### 5.1 COMPARISON

The closest pre-existing research is the supervised learning approach presented in Susaiyah et al. (2021). Although it focuses on insight generation with feedback, it models the problem as a insight scoring problem where each insight is assigned a score using a neural network. Insights were chosen on need-basis to satisfy the accuracy constrains of the system. Whereas, in our work, we focus on showing the single best insight to the user. Although comparison in terms of accuracy could be difficult due to different objectives, we could compare their results with ours in terms of performance drops. In the former technique performance drops were observed a few days into beginning of a new user interest. That is, even after providing feedback on user preference, the system showed insights contradicting to the feedback. However, in our system we do not see such performance drops. In Figure 6, we observe that once our system latches on to a particular user interest, it almost never deviates, indicating a robust insight selection policy. There was however a drop in performance when the user was interested in sleep-duration insights. This could be improved with sufficient training.

On the other hand, it is not easy to make a comparison with other RL or DRL based recommender systems. As presented in Afsar et al. (2021), a lot of effort have been put by the RL community to design RL recommenders in healthcare for clinical decision. However, our framework is dedicated to the improvement of day-to-day individual behaviors, therefore out of the scope of those recommender systems as well as their work is out of our scope. Furthermore, unlike the MIMIC dataset, the ATUS dataset does not include any type of intervention.

### 5.2 EFFORT NEEDED TO ADD A NEW EVENT

The design of the user simulation and DRL-based insight selection allows a very flexible usability for further experiments. As an example, only 3 simple steps would have to be carried if we wanted to add one more activity. First, a new state need to be added to the state machine representing the user and the associated Gaussian transitions computed thanks to the log data of the individual. Then, a new schema should be added to the insight generator in order to compare the lifestyle measurement as described in section 3.1. Additionally, the new measurement could be incorporated in the reward function if it had an impact on the life quality. This also needs to include a dedicated lifestyle improvement factor as explained in section to model how such an insight will affect the behavior of the user.

## 6 CONCLUSION

In this paper, we demonstrated the capabilities of our RL system to recommend insights with the objective of (1) improving the user's life quality and (2) satisfying their interest. From the same observation matrix encoding the history of the selected insights, the policy network is able to reach completely different objectives by a simple modification of the reward function. From an abstract representation of the insights, it is able to understand the impact of the recommendation encoded in the insights' features and efficiently select the most relevant. Furthermore, the random nature of the simulation ensures the robustness of the selection policy, using generic and reusable principles based on very simple assumptions. This framework allows our recommender system to generate and select domain-dependent insights limiting item presentation complexity and time spend for rating while keeping reliability in ratings.

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

# A  APPENDIX

## A.1  EXTRACTION ALGORITHM

---
**Algorithm 1** Extraction algorithm
---
1: **procedure** EXTRACT_AND_BUILD($states$)
2:    **for** $day\ in\ week$ **do**
3:        **for** $s\ in\ states$ **do**
4:            $start, stop, duration \leftarrow$ FILTER_DATA($day, s$)    ▷ All activation times of s on day
5:            $start\_gaussians, stop\_gaussians \leftarrow$ COMPUTE_GAUSSIANS($start, stop$)
6:            ADD_TO_TRANSITION($Idle, s, day, start\_gaussians, 3.5,$ SIZE($duration$))
7:            ADD_TO_TRANSITION($s, Idle, day, stop\_gaussians,$ MAX($duration$), 1)

   **procedure** COMPUTE_GAUSSIANS($start, stop$)
2:    $n\_components \leftarrow 24$
      $start\_gaussians \leftarrow$ FIT_BGMM($start, n\_components$)
4:    $stop\_gaussians \leftarrow$ FIT_BGMM($stop, n\_components$)
      **return** $start\_gaussians, stop\_gaussians$

   **procedure** ADD_TO_TRANSITION($start\_state, end\_state, day, gaussians, horizon, duration$)
      $state\_machine[start\_state][end\_state][day].start\_gaussians \leftarrow gaussians$
3:    $state\_machine[start\_state][end\_state][day].horizon \leftarrow horizon$
      $state\_machine[start\_state][end\_state][day].proportion \leftarrow proportion$
---

## A.2  POLICY NETWORK

As for the user simulation, lots of efforts were put on the environment to keep it as reusable as possible to follow the general framework presented in Figure 1. To code this framework, we developed our solution using the library Stable Baselines 3 (Raffin et al., 2021), again simplifying the reusability of our concepts. However, the main drawback of our standard DRL approach is still the need for a lot of training. For this reason, we decided to use the policy network PPO, described in Schulman et al. (2017), as it was demonstrated to be scalable to large models, allow parallel implementation for training and is more efficient than A2C in terms of sample complexity. The clipped surrogate objective of PPO in equation 8 makes it a perfect fit to avoid instability that could be caused by the sparse reward function we are using for user feedback.

$$L^{CLIP}(\theta) = \mathbf{E}_t[min(r_t(\theta)A_t(\theta),\ clip(r_t(\theta, 1 - \epsilon, 1 + \epsilon)A_t(\theta)))] \tag{8}$$

with $\theta$ the weights of the neural network, $t$ the current time step, $A$ the advantage function and $\epsilon$ a hyperparameter.

## A.3 Additional results

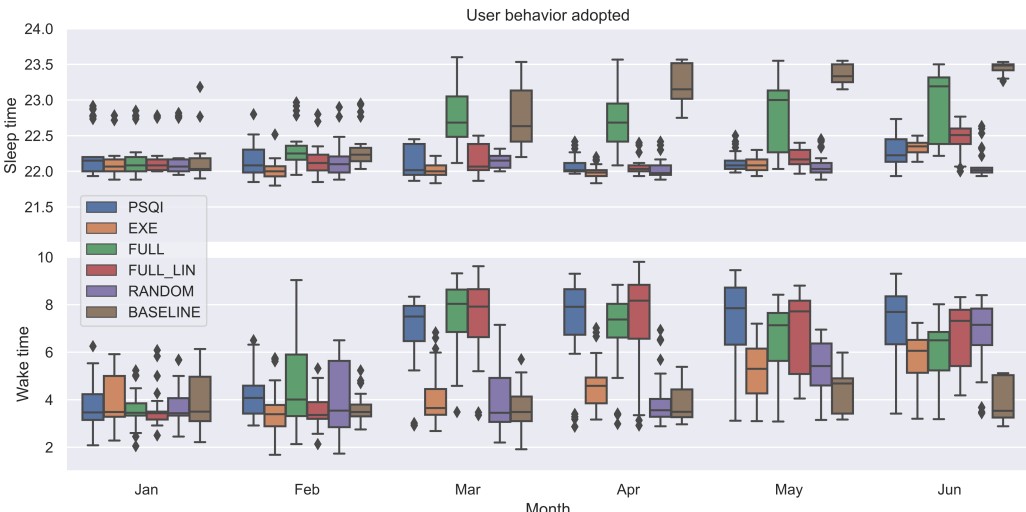

Figure 7: Resulting user's behavior with different insights selection policies during one test simulation.

In the figure 7 is presented how behavior of the user, sleep time and wake time, evolves over the simulation time. On the upper plot we can see that the PSQI, EXE, FULL_LIN and RANDOM policies are selecting enough insights recommending to sleep early in order to keep a reasonable sleep time between 22:00 and 22:30. On the other end, without recommendations about sleep time as for the BASELINE or for FULL, the user is sleeping on average after 23:00. by the end of the simulation. On the lower plot we can see that the PSQI, FULL and FULL_LIN policies are the ones recommending the most insights about wake time as the user is waking up after 06:00 as early as March, when it is only in June for EXE and RANDOM.

## A.4 Future work

We will be deploying this system in patient monitoring and personal health monitoring applications to study and explore insights from patient and personal health. Further work is needed on how to automatically encode the relative impact of each insight on the behavior of the user. Experiments will be conducted both on the relevance of the user simulation system and capabilities of the policy network to adapt to different user behaviors and reward functions. In this study, the same simulation was used during all the training, it would require to train the policy with different users to make it learn how to give good recommendations for different types of lifestyle.

For a matter of simplicity for this preliminary study, only four interests were possible for the user. As the policy was almost perfectly suited for these four interests, we can extrapolate that it would also work if there were more topics of interest with enough training time. Nevertheless, a real user might only be interested in five to ten of those subjects. We believe it would be possible to take this into account by using a weighted random sampling of interests during the training of the policy network, suited to the likelihood of interest of the specific user, instead of a uniform random sampling.

