# OpenReview forum: "Deep Reinforcement Learning based Insight Selection Policy"
_ICLR.cc/2023/Conference — Submitted to ICLR 2023_

### Official Review · Reviewer_a8Lx · 2022-10-20

**Confidence:** 4
**Correctness:** 3
**Technical Novelty And Significance:** 2
**Empirical Novelty And Significance:** 3
**Recommendation:** 5

**Clarity, Quality, Novelty And Reproducibility:**

Clarity:
- I was a bit confused about the example in the first paragraph of section 3.1, it's not very clear what "sleep_period:1 measurement measurement_benchmark: 2" means.
- "Insight generation is performed using the insight generator described in ....", the paper referenced is very recent, so it would be helpful for the reader to get a quick overview of the method so the paper is self contained.
Other than that, I found the paper easy to follow.

Quality & Originality:
The paper is well organized, and contains a detailed description of the methods used and experimental setup.
I also found the work to be original in its use of RL for insight selection.

**Strength And Weaknesses:**

Strengths:
I found the problem and use of RL to be really interesting in this scenario.
The authors explanation of how they modeled user behavior and the decisions made to create the simulator; while the work itself has the potential to be impactful.


Weaknesses:
There are a few key limitations in the work presented:
1 - The user is assumed to only be in one state at a time, but people do more than one thing at a time regularly. For example, it is not uncommon that people eat while traveling or working.

2 - In section 3.2.1, the authors state that  promotion of time Working is greater than Shopping on weekdays, and the other way around on weekends. This really depends on the job, which leads me to think that there's no notion of personalization in the state transitions.

3 - The paper results rely solely on the model used for the simulation, where the reward function has been design to give a clear learning signal. It is still unknown to what extent the insights picked by RL would influence a user's behavior.

**Summary Of The Paper:**

The authors propose a new framework for insights selection in health data based on reinforcement learning. Insights are actionable interpretations of analysis of data that originates from users' behavior.
The paper proposes to create a large list of candidate insights, which are then scored and filtered based on these scores. The RL problem is then formulated as an insight selection problem, where the RL agent learns to pick which insight from the candidates to present to the user, and the reward depends on the topic of the insight selected.

Through a simulation, the authors show that using an RL framework they are able to influence a user behavior to achieve positive outcomes.

**Summary Of The Review:**

Overall, this is an interesting paper with promising results.

However, given that all the evaluation was done in simulation, it is hard to tell what the impact of such a system truly is.
It is already known that for a well defined MDP such as the one in Figure 2a, RL systems will be able to improve their return, so that in itself is not novel. The real question would be if users respond in such a way at this system.

As it stands, to me this looks like a promising work that's missing that key component in the evaluation.

---

> ### Author Response · Authors · 2022-11-19
> **Re-written parts to add more clarity on insight generation and MDP representation. Addressed all other questions.**
>
> Thank you very much for your time in reviewing this paper. Please find below our responses.
>
> 	Comment 1: The user is assumed to only be in one state at a time, but people do more than one thing at a time regularly. For example, it is not uncommon that people eat while traveling or working.
>
>     Response: As a solution, a single state can combine multiple activities that co-occur, such as eating-travelling and eating-working. The only downside is that it loses the statistical strength if the dataset is of small size. However, this would be taken care of by the proportionality factor described in Section 3.3.1.
>
> 	Comment 2:  In section 3.2.1, the authors state that promotion of time Working is greater than Shopping on weekdays, and the other way around on weekends. This really depends on the job, which leads me to think that there's no notion of personalization in the state transitions.
>
>     Response: This was stated as an example to explain the proportionality constrain. It is automatically computed from the data distributions and helps to generate realistic likelihoods of events. This has been rewritten in the above section for better clarity.
>
>     Comment 3: The paper results rely solely on the model used for the simulation, where the reward function has been design to give a clear learning signal. It is still unknown to what extent the insights picked by RL would influence a user's behavior.
>
>     Response: In our study, we assumed that the user’s behavior will change by a behavior-change factor explained in Section 3.4.2. However, it helps us to observe the usefulness of using a RL algorithm in such a dynamically constrained use case. As a first step toward the evaluation of this novel approach, the results are promising. In the future, experiments with dynamically changing factors of behavior-change based on actual patient will be conducted in order to determine what is their impact on the learning process.
>
> 	Comment 4: I was a bit confused about the example in the first paragraph of section 3.1, it's not very clear what "sleep_period:1 measurement measurement_benchmark: 2" means. "Insight generation is performed using the insight generator described in ....", the paper referenced is very recent, so it would be helpful for the reader to get a quick overview of the method so the paper is self contained.
>
> 	Response: We have rewritten Section 3.1 in order to give an example for each of those schemas. Therefore, we hope that it is now easier to understand how the insight generator is working. Even thought the referenced paper is very recent, all information essential to the understanding of our RL framework can be found in the previously mentioned section. As the novelty of this work is based on the user simulation and RL objectives, the reader eager to understand in a more detailed way how the generation is performed will be able to read the mentioned paper.
>
> 	Comment 5: Given that all the evaluation was done in simulation, it is hard to tell what the impact of such a system truly is. It is already known that for a well defined MDP (missing out the IG)such as the one in Figure 2a, RL systems will be able to improve their return, so that in itself is not novel. The real question would be if users respond in such a way at this system. As it stands, to me this looks like a promising work that's missing that key component in the evaluation.
>
> 	Response: There seems to have been a slight misunderstanding about the Figure 2a as what is represented is the simulated user as a state machine with Gaussian mixture model based transition, it is therefore not an MDP. Furthermore, this information is never given to the RL policy in any way as its observation is based on the history of previously selected insight as explained in Section 3.4.1. Therefore, despite the model of the user being rather simple (i.e., only 8 states), the selection problem is greatly more difficult to model under a formal MDP. Following this comment, we have rewritten several parts of the paper in order to have more clarity on the MDP formulation of the problem. Also, as explained in the response to your Comment 3, human evaluation will be the next development step of this framework.
>
> Overall Response:
> We once again thank the reviewer for their valuable comments. We have reflected on these remarks with the inclusion of additional information in the paper. We hope it addresses the issues adequately. We would like to hear from you on your opinion.

---

> > ### Comment · Reviewer_a8Lx · 2022-11-21
> > **Response to authors**
> >
> > Thank you very much for editing the paper to address some of my comments.
> > I think you did a great job of clarifying some misunderstandings in section 3, that extra explanation makes the design a lot easier to follow.
> >
> > Overall, I like this paper and the direction where it's going, but I still believe that since the key point is to influence users' behavior and the contribution of this work is mostly empirical, an evaluation with real users is necessary.
> > My reasoning is that if the results seen in simulation cannot be replicated in a real scenario, the value of the contribution would be rather limited.

---

> > > ### Author Response · Authors · 2022-11-29
> > > **Response to reviewers**
> > >
> > > Dear Reviewer,
> > >
> > > Thank you very much for your comment. As far as behaviour change is concerned, performing human evaluations is of utmost importance. However, such an evaluation is also highly regulated due to GDPR, and it would be dangerous to directly test it on users without simulated experiments like those presented in the paper.
> > >
> > > Besides the lifestyle parameters discussed in this paper, aspects such as psychological and physical impact should also be considered. By presenting this framework, we hope to stimulate further research along these lines and allow researchers from different backgrounds to contribute to its development. Based on the simulation results, we believe DRL shows great promise, and the forum is an ideal place to showcase our work.

---

### Official Review · Reviewer_XoQA · 2022-10-25

**Confidence:** 4
**Correctness:** 2
**Technical Novelty And Significance:** 1
**Empirical Novelty And Significance:** 1
**Recommendation:** 3

**Clarity, Quality, Novelty And Reproducibility:**

I have provided detailed comments related to clarity, quality, novelty, and reproducibility in the weaknesses section.

**Strength And Weaknesses:**

Strengthes:
1. The research topic of this paper is personal health, which is meaningful and socially impactful.
2. The authors provide a practical RL framework to comprehend user preferences.

Weaknesses:
1. The presentation of this paper is not good. Readers cannot quickly get the main contributions and novelties.
2. The novelty of this paper is limited. All the key concepts and techniques have existed in a lot of literature.
3. This paper's technical depth is limited. The writers did not derive an adequate research concept from the issue. Numerous paragraphs are used to explain data extraction and processing procedures. They are, however, too insignificant. This work identifies a suitable application subject, however, the authors should propose their own contributions in addition to applying RL to the problem.
4. The authors didn't provide code and data links to improve their reproducibility.
5. Experimental design is limited. The authors should design more case studies and ablation studies to illustrate the effectiveness.

**Summary Of The Paper:**

Summary: The growth of sensors and Internet of Things techniques has made it possible to collect an increasing amount of data, which can then be analyzed for its patterns in order to enhance the applications that are linked with those patterns, especially in the field of personal health, where a large amount of data may be applied to the understanding of users' living behaviors and the indirect improvement of their way of life. Systems that are able to recognize these patterns and translate them into a text format that is easily readable are referred to as insight generators. The authors of this study present a unique reinforcement learning (RL) framework for insight selection. This framework has the potential to be utilized in order to both evaluate the authors' lifestyle quality and capture the authors' usage interests. Experiments have shown that RL has the potential to improve the selection of insights toward a number of different pre-defined goals.



**Summary Of The Review:**

This paper studies an interesting research topic and proposes a practical framework. But it has the following limitations:
1. The presentation of this paper is bad.
2. The novelty of this paper is limited.
3. The technical depth of this paper is restricted.
4. The reproducibility of this paper is bad.
5. The experimental design of this paper is incomplete.
Based on above limitations, I prefer to reject this paper.

---

> ### Author Response · Authors · 2022-11-19
> **Rewritten parts to add more clarity and incorporated other comments.**
>
> Thank you very much for your time in reviewing this paper. Please find below our responses.
>
> 	Comment 1: The presentation of this paper is not good. Readers cannot quickly get the main contributions and novelties.
>
> 	Response: We have taken this into great consideration and rewritten several parts to make it concise and informative at the same time.
>
> 	Comment 2: The novelty of this paper is limited. All the key concepts and techniques have existed in a lot of literature.
>
> 	Response: The main novelty in the first draft was not clearly mentioned and has been addressed in the revision. To summarize, our framework specifically focuses on health behavior insights generated using a non-parametric statistical model based generator additionally. There are no works in literature that addresses such an insight selection policy using DRL. Additionally, we have a novel objective of choosing insights that are both useful and preferable by the user.
>
> 	Comment 3: This paper's technical depth is limited. The writers did not derive an adequate research concept from the issue. Numerous paragraphs are used to explain data extraction and processing procedures. They are, however, too insignificant. This work identifies a suitable application subject, however, the authors should propose their own contributions in addition to applying RL to the problem.
>
> 	Response: We have rewritten several parts to address this. Also, we have taken more care to highlight the technical contribution. We believe that the technique we introduce to simulate behavior and behavior change would be a valuable contribution. Additionally, we identified suitable reward mechanisms to enforce behavior change. This could also be further substantiated using user studies.
>
> 	Comment 4: The authors didn't provide code and data links to improve their reproducibility.
>
>     Response: The code for the insight generator will be published in a separate paper.
>
> 	Comment 5: Experimental design is limited. The authors should design more case studies and ablation studies to illustrate the effectiveness.
>
> 	Response: Our objective was limited to the scope to find to what extent RL can help in a simulated environment. The result indicate promising performance for different types of modeling the models such as : topic selection, insight selection, beneficial insights objective and preferable insight objective. However, it does lack a user evaluation, which would be a promising follow-up case study to the experiment.
>
> 	Overall Response:
> 	We once again thank the reviewer for their valuable comments. We have reflected on these remarks with the inclusion of additional discussion and appropriate rewriting to highlight the key contributions.

---

### Official Review · Reviewer_p6Dn · 2022-10-31

**Confidence:** 4
**Correctness:** 3
**Technical Novelty And Significance:** 3
**Empirical Novelty And Significance:** 2
**Recommendation:** 5

**Clarity, Quality, Novelty And Reproducibility:**

The paper is well-writhing and motivated. The main contribution is the use of RL algorithm in the insight selection problem.


**Strength And Weaknesses:**

Strengths:
1. It is an interesting idea to generate and select insights using the reinforcement learning diagram.
2. The paper is well organized, with a good hierarchical structure and clear chapter headings.

Weaknesses:
1. This paper concentrates on two kinds of insights, insights that are appreciated by the user and insights that are beneficial to their life quality; what is the relationship between them? The article does not clearly explain.
2. Since the American Time Use Survey (ATUS) 2003-2020 dataset exists, can the supervised learning method and the reinforcement learning framework proposed in this paper be compared with the experimental results?
3. If it is difficult to compare with a supervised learning framework, at least compare with a framework that is also modeled by MDP, such as works in the survey by Afsar et al. (2021).
4. In this paper, real life is modeled in the offline dataset ATUS by state machines. If a state outside of the dataset emerges in a real-world application, how does the insight selection network make decisions?
5. Some writing errors, such as " Those subjects have been selected to tell to the policy network about what measurement is the insight, if it compares it to the benchmark value and to which day of the week it refers to.", and "On figure 6 is presented the behavior of the policy network after 12.000.000 steps of training." and so on.

**Summary Of The Paper:**

This work provides a reinforcement learning solution for the insight selection problem and use two experiments to verify the feasibility of the proposed framework. The main claimed contribution is that the framework can provide insights that are both relevant to user preferences and improve users' healthcare. Preliminary experimental result on the American Time Use Survey 2003-2020 shows that the proposed RL solution outperforms insights from multiple pre-defined objectives.

**Summary Of The Review:**

The article provides a reinforcement learning solution to the insight selection problem, and it would be nice to have a more detailed experimental comparisons, analyses, and discussion to highlight the technical contribution of this work besides only using the RL algorithm in a new task.

---

> ### Author Response · Authors · 2022-11-19
> **Included discussion talking about the comparability and address all other questions.**
>
> Thank you very much for your time in reviewing this paper. Please find below our responses.
>
> 	Comment 1: This paper concentrates on two kinds of insights, insights that are appreciated by the user and insights that are beneficial to their life quality; what is the relationship between them? The article does not clearly explain.
>
> 	Response:  This is mainly to address the users who sometime not prefer to see beneficial insights for various reasons. We have added more details in section 4.2.
>
> 	Comment 2:  Since the American Time Use Survey (ATUS) 2003-2020 dataset exists, can the supervised learning method and the reinforcement learning framework proposed in this paper be compared with the experimental results?
>
>     Response: The main reason why we couldn't use the ATUS to validate the experimental results is the unavailability of interventional data in which users provided feedbacks to insights. This is further explained in the Discussion (Section 5.1) and This was one of the motivation to develop the Simulation environment as mentioned in Related Work (Section 2). Also, it cannot be compared fully to the supervised learning approach due to difference in the training objectives as explained in Section 5.1. However, we have presented qualitative comparison in the same section.
>
> 	Comment 3: If it is difficult to compare with a supervised learning framework, at least compare with a framework that is also modeled by MDP, such as works in the survey by Afsar et al. (2021).
>
>     Response: A detailed explanation on comparability is provided in Section 5.1. The main limitation is that our environment is not an MDP due to the introduction of non-parametric insight scoring algorithm that picks insights from the data and also the data simulator used in our framework is based on Gaussian mixture models.
>
> 	Comment 4: In this paper, real life is modeled in the offline dataset ATUS by state machines. If a state outside of the dataset emerges in a real-world application, how does the insight selection network make decisions?
>
> 	Response: It is indeed necessary to inform the system about the new state. A detailed step by step process to add a new state is now presented in Section 5.2
>
> 	Comment 5: Some writing errors, such as " Those subjects have been selected to tell to the policy network about what measurement is the insight, if it compares it to the benchmark value and to which day of the week it refers to.", and "On figure 6 is presented the behavior of the policy network after 12.000.000 steps of training." and so on.
>
>     Response: Thank you for pointing out. These have been rewritten for clarity.
>
> Overall Response:
> We once again thank the reviewer for their valuable comments. We have reflected on these remarks with the inclusion of additional discussion and appropriate rewriting to highlight the key contributions. We hope it addresses the issues adequately. We would like to hear from you on your opinion.

---

### Decision · Program_Chairs · 2023-01-20

**Decision:**

Reject

**Justification For Why Not Higher Score:**

Lack of novelty, simplistic simulation, no real-world data.

**Justification For Why Not Lower Score:**

N/A

**Metareview: Summary, Strengths And Weaknesses:**

Insight selection is an uncommon and potentially novel application topic. This paper is not very convincing, however. Most importantly, no real-world data is involved, only a simulation that is rather simplistic and limited in many ways. It is not clear what we can learn from this. Additionally, the actual RL method is not very novel.